# Empathy, Burnout, and Attitudes Toward Patients with Mental Disorders Among Mental Health Nurse Residents in Spain: A Cross-Sectional Study

**DOI:** 10.3390/nursrep15110381

**Published:** 2025-10-27

**Authors:** Daniel Román-Sánchez, Anna Bocchino, José Luis Palazón-Fernández, Concepción Mata-Pérez, Alberto Cruz-Barrientos, José Manuel de la Fuente Rodríguez, Juan Carlos Paramio-Cuevas

**Affiliations:** Salus Infirmorum Nursing Center, University of Cádiz, 11001 Cádiz, Spain; daniel.roman@ca.uca.es (D.R.-S.); anna.bocchino@ca.uca.es (A.B.); conchi.mata@ca.uca.es (C.M.-P.); alberto.cruzba@ca.uca.es (A.C.-B.); jose.defuente@ca.uca.es (J.M.d.l.F.R.); juancarlos.paramio@ca.uca.es (J.C.P.-C.)

**Keywords:** empathy, burnout, mental health nursing, nursing residency, stigma, Spain

## Abstract

**Background/Objectives:** Mental health nurses are among the health professionals who show the highest levels of empathy and the most positive attitudes towards patients with mental disorders. Nevertheless, burnout is a prevalent issue among these professionals, as well as throughout their training as residents. The objective of this study is to examine the relationship between empathy, burnout and attitudes towards patients with mental disorders among mental health nurse residents in Spain. **Methods**: A cross-sectional descriptive design was employed with a sample of 214 resident nurses engaged in specialty training in mental health teaching units in Spain. The sample was selected purposely. The Jefferson Empathy Scale, the Maslach Burnout Inventory and the Community Attitudes Towards Mental Illness Inventory were employed to quantify the study variables. **Results**: The study sample included 214 residents, composed of 46 males (21.5%) and 168 females (78.5%), with an overall median age of 27.00 years (IQR: 25.00–29.00). The internal reliability of the instruments was high (JSE α = 0.834, MBI α = 0.821, CAMI-CMH α = 0.851). Our findings indicate that empathy during residency is associated with an enhancement in positive attitudes towards patients with mental disorders, and a reduction in the associated stigma (JSE vs. CAMI-SR: ρ = −0.210, *p* < 0.01). However, empathy did not act as a protective factor against burnout in the study sample, showing a moderate positive correlation with emotional exhaustion (JSE vs. MBI-EE: ρ = 0.481, *p* < 0.01). **Conclusions**: Our findings suggest that although empathy does not necessarily act as a protective factor against burnout, it may favour more positive and less stigmatising attitudes towards people with mental disorders. Future research is needed to further investigate the complex interplay between emotional competencies and work-related stress in this context.

## 1. Introduction

The connection established between nurses and patients is considered a central component of nursing practice and is usually referred to as the therapeutic relationship [1,2]. The broadening of nursing competencies has progressively modified the nature of this relationship [3,4]. Within this context, empathy is regarded as one of its essential foundations [5]. It is defined as a multidimensional construct [6] that allows professionals to recognise the perspectives, concerns, and lived experiences of others without losing their own self-awareness [7], and it is influenced by both education and professional practice [8].

Research consistently demonstrates the multiple benefits that empathy brings to healthcare. Among these are the improvement of care quality, stronger bonds between patients and professionals, better treatment adherence, enhanced clinical results, and higher satisfaction on the part of both patients and healthcare workers [9,10]. Empathy is, in fact, the interpersonal quality most valued by healthcare users and professionals alike [11]. This is why it has been the focus of numerous studies, aimed at strengthening nurses’ performance and their ability to interact effectively with patients [9,12].

According to Petrucci et al. [13], nursing is the healthcare field in which empathy plays the most significant role in professional–patient relationships. For that reason, it is identified as a key communication competence in this profession [5], which explains the large number of studies devoted to analysing empathy in nursing practice [14].

Burnout also plays an important role in shaping therapeutic relationships. This syndrome is a major stress factor for professionals and has been associated with decreased patient satisfaction [15]. Variables such as years of experience, stress, time limitations, interactions with colleagues, and the heavy demands of care provision can make empathic communication more difficult, thereby increasing the risk of burnout [16]. This condition is typically characterised by depersonalisation, emotional exhaustion, and diminished personal accomplishment, which obstruct effective communication and diminish the ability to understand patients’ needs [17]. On the other hand, high levels of empathy may act as a protective element against burnout [18].

A further challenge in healthcare practice is stigma towards people with mental disorders. This stigma becomes a powerful barrier to their integration in social, occupational, and family life [19,20]. Evidence shows that, in healthcare environments, professionals may also display stigmatising attitudes towards psychiatric patients [21], jeopardising the development of effective therapeutic relationships. Nonetheless, mental health nurses, along with psychiatrists, are the professionals who demonstrate the highest empathy and the most supportive attitudes toward these patients [22,23]. In this regard, empathy in mental health nursing, as highlighted by Tippin and Maranzan [24], emerges as a crucial factor in diminishing stigma and facilitating more positive therapeutic engagement.

In Spain, nurses specialising in mental healthcare receive training in therapeutic relationships [25], which includes the development of empathy. This is of particular importance given the vulnerability of patients with mental disorders [22,23,24] who often have to live with the prejudice and stigma associated with their condition, causing a significant negative impact on their quality of life.

This study is also aligned with international frameworks promoting compassionate, person-centred care, such as the WHO Comprehensive Mental Health Action Plan (2020–2030) and the International Council of Nurses (ICN) standards for nursing education [26,27], which emphasise empathy and emotional competence as core components of professional development. Furthermore, while previous studies have examined empathy, burnout, and attitudes in healthcare professionals, many have employed heterogeneous methodologies or lacked cultural contextualization. These gaps highlight the need for research addressing these variables simultaneously within specific educational and cultural frameworks, such as that of Spanish mental health nursing.

The study of empathy and its associated factors (burnout and attitude towards the mentally ill) is fundamental to understand and enhance the therapeutic relationship between nurses and patients with mental disorders [5,28]. Despite this importance, there is still a limited number of studies exploring empathy levels among residents in health science fields, with most existing investigations focusing predominantly on medical specialties [29,30]. Research carried out on empathy among residents and practicing nurses from different specialties has largely been situated in emergency and primary care contexts [31,32].

In Spain, the training of Mental Health Nurse Residents (EIR) follows a two-year national residency programme regulated by the Spanish Ministry of Health, which combines clinical practice in psychiatric units, community mental health centres, and addiction services. The educational structure emphasises technical and therapeutic competencies but often provides limited formal training in emotional regulation or empathy development. Moreover, cultural factors such as persistent stigma toward mental illness in Spanish society may shape both professional attitudes and emotional responses to patients. Previous studies have indicated that social distance and negative stereotypes regarding mental disorders remain prevalent in the Spanish context, which could influence nurses’ empathic engagement and attitudes toward these patients. Considering these educational and sociocultural aspects provides a relevant background for interpreting the interaction between empathy, burnout, and attitudes in this population.

Therefore, the aim of this study is to describe levels of empathy, burnout, and attitudes towards mental health patients in a group of resident nurses specialising in mental health. Additionally, it seeks to examine the relationship of these variables with a set of socio-demographic characteristics.

The study hypothesises that residents with greater empathy will exhibit lower levels of burnout—reflected in reduced emotional exhaustion, diminished depersonalisation, and stronger personal accomplishment—as well as more favourable attitudes towards patients with mental disorders, expressed through lower scores in authoritarianism and social restrictiveness, and higher levels of benevolence and community-oriented ideology in mental health.

## 2. Materials and Methods

### 2.1. Study Design and Participants

This study is a descriptive, observational, cross-sectional investigation carried out within the teaching units of the Mental Health Nursing specialty of the Spanish National Health System. In relation to the institutional characteristics in the training plan of the Internal Nurse Resident (INR) specialty, the process is standardised across all autonomous communities in Spain, aiming for a homogeneous training pathway throughout the Spanish territory. The institutional settings for the Mental Health Nursing specialty are primarily based on the existing homogeneity within the psychiatric units of the clinical management areas of the hospitals. The study population comprised all internal nurse residents (INR) of the mental health nursing specialisation who met the eligibility criteria. A priori power analysis using GPower indicated that a minimum sample size of 197 participants was required for an effect size of 0.25 with a power of 0.95 and α = 0.05. The final sample of 214 participants exceeded this threshold. Since the residency for the Mental Health Nursing speciality in Spain lasts two years, we stratified the sample based on the year of residency. Participants were recruited from different teaching units of the Spanish National Health System, distributed across several regions. The inclusion criteria were active enrolment as a Mental Health Nursing Resident during the 2018–2019 academic year, being an adult (over 18), and providing informed consent, while exclusion criteria included residents on leave, those with previous psychiatric specialisation, or incomplete survey responses. A purposive, non-probabilistic, non-discriminatory, exponential snowball sampling method [33] was employed in order to recruit eligible subjects from different regions of Spain. This sampling strategy enables access to hard-to-reach populations and facilitates the expansion of the sample size. The sampling process begins with the identification of initial participants, or “seeds,” who are excluded from subsequent analyses. These seeds serve as recruitment agents, inviting additional participants from within the same population, thereby promoting a snowball effect [34].

In this study, the target population comprised 438 internal residents specialising in Mental Health Nursing within the Spanish National Health System, as defined by the official allocation of residency positions in Spain for the years 2018 and 2019. Recruitment was supported by collaborators from the various teaching units in Mental Health Nursing across the Spanish autonomous communities, who acted as the designated “seeds.” The study instruments were disseminated through these seeds, ensuring the broadest possible distribution among mental health nurse residents nationwide. To obtain the data, the researchers made personal contact with the collaborators of the teaching units in which the procedure had to be carried out.

### 2.2. Instruments and Variables

Ad hoc demographic questionnaire: It contained questions related to age, gender and residency year.

-Empathy

Empathy was measured using the Spanish adaptation of the Jefferson Scale of Empathy (JSE) [35], comprising 20 items pertaining to empathic responsiveness in healthcare contexts. Ten items are worded in a positive manner, while the remaining ten are worded in a negative manner, thus preventing unreflective responses. Each item is assigned a score on a 7-point Likert scale, ranging from 1 (indicating strong disagreement) to 7 (indicating strong agreement). In order to ensure consistency and comparability, scores for negative items are reversed, so that individuals responding ‘strongly disagree’ are awarded 7 points. The JSE is composed of three dimensions: perspective taking (JSE-PT) (items 2, 4, 5, 9, 10, 13, 15, 16, 17, 20), reflects cognitive empathy; compassionate care (JSE-CC) (items 1, 8, 11, 12, 14, 19), is the emotional empathy; and finally, standing in the patient’s shoes (JSE-SPS) (items 3, 6, 7, 18), makes up a residual dimension referred to the adoption of the patient’s perspective. The score for each subscale is calculated by summing the points obtained on all items. The total scores range from 20 to 140 points, with higher scores indicating greater levels of empathy. In the adaptation to Spanish, the instrument demonstrated high internal consistency and a reliability of 0.82. The JSE is currently the most widely used instrument for measuring this variable [13].

-Burnout

The level of burnout among healthcare professionals was measured using the Spanish version of the Maslach Burnout Inventory (MBI). The scale comprises 22 items presented in the form of statements regarding professionals’ feelings and attitudes towards their work and patients. The respondents are asked to indicate the frequency with which they experience these feelings on a 7-point Likert scale, ranging from 0 (never) to 6 (every day). The MBI comprises three dimensions: emotional exhaustion (MBI-EE) (items 1, 2, 3, 6, 8, 13, 14, 16, 20), feeling emotionally drained by work; depersonalization (MBI-DP) (items 5, 10, 11, 15, 22), becoming emotionally detached due to job demands; and personal accomplishment (MBI-PA) (items 4, 7, 9, 12, 17, 18, 19, 21), having a positive impact on others through work. Each subscale score is calculated by totalling the points awarded for all items. However, items phrased negatively are scored inversely. According to Seisdedos [36], high scores on MBI-EE and MBI-DP and low scores on MBI-PA indicate high levels of burnout. The Spanish adaptation of the scale shows high internal consistency and reliability, with Cronbach’s Alpha values of 0.90 for emotional exhaustion, 0.79 for depersonalization, and 0.71 for personal accomplishment. The original MBI [37] is currently considered the most valued and widely used instrument for measuring burnout in national and international studies.

-Attitudes toward individuals with mental illness

Attitudes towards mentally ill patients were measured using the Spanish version of the Community Attitudes towards Mental Illness (CAMI) inventory [38]. The scale has been used in a variety of populations, including nurses, psychiatrists, family members, and the general population. The scale consists of 40 items rated on a 5-point Likert scale ranging from 1 (strongly agree) to 5 (strongly disagree) It is made up of four factors: authoritarianism (CAMI-A) (items 1, 5, 9, 13, 17, 21, 25, 29, 33, 37), designed to assess community attitudes towards the mentally ill; benevolence (CAMI-B) (items 2, 6, 10, 14, 18, 22, 26, 30, 34, 38), which examines welcoming attitudes towards patients; social restrictiveness (CAMI-SR) (items 3, 7, 11, 15, 19, 23, 27, 31, 35, 39), which assesses the belief that people with mental disorders are a danger to society; and community mental health ideology (CAMI-CMH) (items 4, 8, 12, 16, 20, 24, 28, 32, 36, 40), which determines beliefs related to the integration of people with mental disorders into society. These factors are composed of 10 statements that address views on the treatment and care of individuals with a mental disorder. Each factor comprises five positively worded items and five negatively worded items. The score for each factor is calculated by adding together the points awarded to all items. However, items phrased in a negative manner are scored inversely. For instance, if a respondent assigns a score of 5 to an item expressed in a negative way, the score is considered to be 1.

A high score on the CAMI-B and CAMI-CMH, coupled with a low score on the CAMI-A and CAMI-SR, suggests a positive attitude towards mental health patients. Following the adaptation to Spanish, the instrument demonstrated an internal consistency of 0.86.

### 2.3. Procedure

Data were collected by the research team through a self-administered online questionnaire designed on the SurveyMonkey platform. The survey link was distributed via institutional email to all Mental Health Nursing Teaching Units within the Spanish National Health System, inviting the voluntary participation of residents. Local teaching coordinators collaborated only in disseminating the link and did not have access to the responses. The data collection process took place between May and November 2019, and a total of 214 mental health nurse residents (109 first-year and 105 second-year residents) completed the anonymous questionnaire, which included sociodemographic variables (age, sex, and year of residency), as well as validated measures of empathy, burnout, and attitudes toward patients with mental illness. Participants provided informed consent electronically prior to participation. To ensure data quality and confidentiality, only one submission per device was permitted, IP addresses were checked to prevent duplicates, and responses were automatically coded and anonymized before analysis.

### 2.4. Statistical Analysis

Qualitative variables (sex) were described using absolute and relative frequencies. Quantitative variables (age, JSE, MBI, and CAMI) were expressed as medians and interquartile ranges (IQR), as they showed substantial deviations from the normal distribution according to the Kolmogorov–Smirnov test. The reliability of the JSE, MBI, and CAMI scales was assessed using Cronbach’s alpha. To evaluate the performance and quality of the items in the different scales we performed an item analysis that included item-total correlations and Alpha-if-item-deleted statistics. The correlation between the JSE, MBI, and CAMI scores and between these and sociodemographic variables was assessed using Spearman’s rank correlation coefficient. Comparisons of quantitative variables between two independent groups were performed using the Mann–Whitney U test. All tests were two-tailed, setting the statistical significance threshold at 0.05. The type 1 family-wise error rate was controlled at 0.05 by the Holm–Bonferroni correction [39]. There was only 0.03% of missing data and they were deleted in a pairwise fashion. All analyses were performed using IBM’s SPSS© (version 27.0) software (IBM Corp., Armonk, NY, USA).

### 2.5. Ethical Considerations

This research was approved by the Research Ethics Committee of Cádiz, attached to the Andalusian Health Service (approval date: April 2019). The study was conducted in accordance with the Declaration of Helsinki (2013, Seventh Revision, 64th Meeting, Fortaleza) and complied with the general principles of bioethics, current Spanish health research regulations, and Organic Law 3/2018 of December 5th on Personal Data Protection and Guarantee of Digital Rights. Prior to data collection, the participants were informed about the objectives of the study and that their participation was completely voluntary so they could leave the study at any time if they did not feel comfortable for any reason. It was also emphasised that the responses were anonymous and confidential. Only the research team had access to the data collected. The academic institutions granted approval for this study.

## 3. Results

The demographic characteristics and the scores for empathy, burnout and attitudes toward the mentally ill of the 214 study participants are shown in Table 1, Table 2, Table 3 and Table 4, and the correlations between the studied variables are shown in Table 5.

The sample consisted of 109 first year residents and 105 second and final year residents. The majority of participants were female in the overall sample (78.5%), as well as in the first (74.3%) and second year (82.9%) of residency. The median age was 27 years (IQR 25–29) (Table 1).

**Table 1 nursrep-15-00381-t001:** Descriptive and demographic data of mental health nurse residents in Spain.

Variable	Total Sample N = 214	R1 N = 109	R2 N = 105
Sex			
Male	46 (21.5%)	28 (25.7%)	18 (17.1%)
Female	168 (78.5%)	81 (74.3%)	87 (82.9%)
Age	27.00 (25.00–29.00)	26.00 (24.00–29.00)	27.00 (25.50–29.50)
Male	25.50 (24.00–29.00)	25.00 (24.00–27.00)	28.00 (25.75–30.00)
Female	27.00 (25.00–29.00)	27.00 (24.00–29.00)	27.00 (27.00–29.00)

Median (interquartile range) for quantitative variables, frequencies (%) for qualitative variables. R1 = first year of residency, R2 = second year of residency.

In this study, the reliability coefficients of the instruments used showed acceptable internal consistency, ranging from 0.70 to 0.90 [39], with the exception of the total score of the JSPE-SPS and the MBI-DP subscale. Cronbach’s alpha for the CAMI-A scale indicated a relatively low level of reliability (α = 0.603). Item-level analysis revealed that most items contributed positively to the internal consistency of the scale, as their removal would have reduced the alpha coefficient. Conversely, the exclusion of items 13 and 29 increased the overall alpha value, as they showed a negative correlation with the total score. Therefore, it was decided to remove these two items from subsequent analyses.

The reliability analysis showed adequate internal consistency for all instruments and their corresponding subscales, with Cronbach’s alpha coefficients within acceptable ranges (α = 0.60–0.89), as presented in Table 2, Table 3 and Table 4.

### 3.1. Empathy

Table 2 shows the results obtained for total empathy and its dimensions. In all of them, statistically significant differences (*p* < 0.05) were observed throughout the years of residence, being greater in the second year both in general and by sex, except in the “compassionate care” dimension where no differences were observed in men (*p* > 0.05).

**Table 2 nursrep-15-00381-t002:** Association between empathy and its dimensions, sex and years of residence of mental health nurse residents in Spain.

	JSPE (α = 0.870)			
Variable	Whole Sample	R1	R2	*p*-Value
	117.00 (107.00–123.00)	111.00 (105.00–118.50)	121.00 (115.00–125.50)	0.000
Sex				
Male	109.50 (105.00–115.00)	106.00 (101.00–109.75)	115.00 (112.50–120.75)	0.000
Female	119.00 (111.00–124.00)	112.00 (106.00–120.00)	122.00 (117.00–126.00)	0.000
*p*-Value	0.000	0.001	0.04	
	**JSPE-SPS (α = 0.602)**			
	12.00 (10.00–12.00)	11.00 (9.00–12.00)	12.00 (12.00–13.00)	0.000
Sex				
Male	12.00 (10.00–12.00)	11.00 (10.00–12.00)	12.00 (12.00–13.00)	0.000
Female	12.00 (10.00–12.00)	11.00 (9.00–12.00)	12.00 (12.00–13.00)	0.000
*p*-Value	0.882	0.324	0.612	
	**JSPE-CC (α = 0.728**)			
	45.00 (42.00–49.00)	44.00 (40.00–46.75)	46.00 (44.27–49.75)	0.000
Sex				
Male	42.50 (40.00–45.00)	41.00 (39.00–44.75)	44.50 (41.00–46.00)	0.056
Female	46.00 (43.00–50.00)	44.00 (40.25–49.0)	47.00 (45.00–51.00)	0.000
*p*-Value	0.000	0.015	0.000	
	**JSPE-PT (α = 0.826)**			
	60.00 (55.00–63.00)	56.00 (54.00–61.00)	63.00 (59.00–64.75)	0.000
Sex				
Male	56.00 (53.00–59.25)	54.00 (50.25–55.75)	59.00 (57.00–63.25)	0.000
Female	61.00 (56.00–64.00)	58.00 (54.00–61.00)	63.00 (59.75–65.00)	0.000
*p*-Value	0.000	0.000	0.061	

Values are medians and interquartile ranges (in brackets). R1 = first year of residency, R2 = second year of residency.

In relation to the comparison by sex, there were only significant differences (*p* < 0.05) between the sexes in general and over the years in total empathy and in the dimension “compassionate care”, where women had higher scores. In the “perspective taking” dimension these differences (*p* < 0.05) were only observed in general and in the first year of residence where women again reached higher values than men (*p* > 0.05) and, finally, in the “standing in the patient’s shoes” dimension there were no significant differences (*p* > 0.05) between men and women.

A positive correlation (*p* < 0.05) was observed between total empathy and its dimension “perspective taking” with all the study variables, with the exception of the “social restrictiveness” and “authoritarianism” dimensions of attitudes towards mentally ill patients, where a negative correlation (*p* < 0.05) was observed (Table 5). These previous correlations were maintained except that, in the dimension “compassionate care” there was no statistically significant correlation (*p* > 0.05) with the total CAMI and in the dimension “standing in the patient’s shoes” there was no statistically significant correlation (*p* > 0.05) with the total burnout and its dimensions “emotional exhaustion” and “depersonalization” (Table 5).

### 3.2. Burnout

Table 3 shows the results for overall burnout and its corresponding dimensions. Among male participants, a statistically significant increase (*p* < 0.05) was observed over the years in both total burnout scores and the “personal accomplishment” dimension. On the other hand, in the “depersonalization” dimension, a statistically significant increase (*p* < 0.05) was observed both in general and in females, which increased its values. Finally, the dimension “emotional exhaustion” showed no differences between the years of residence.

In relation to the comparison by sex, both total burnout and the “emotional exhaustion” dimension were significantly higher (*p* < 0.05) in women both in general and in the first year. However, no sex differences were observed in the “depersonalization” and “personal accomplishment” dimensions.

**Table 3 nursrep-15-00381-t003:** Relationship between empathy and its dimensions, gender and years of residence of mental health professionals in Spain.

	MBI (α = 0.821)			
Variable	Whole Sample	R1	R2	*p*-Value
	41.00 (38.00–50.00)	41.00 (37.00–48.50)	42.00 (39.00–53.50)	0.085
Sex				
Male	40.00 (36.00–43.00)	38.00 (35.00–43.00)	41.50 (39.00–44.00)	0.037
Female	42.00 (39.00–54.75)	42.00 (38.50–52.50)	43.00 (39.00–55.00)	0.511
*p*-Value	0.005	0.014	0.378	
	**BURNOUT-EE (α = 0.894)**			
	2.00 (0.00–10.25)	1.00 (0.00–9.00)	3.00 (0.00–11.00)	0.131
Sex				
Male	1.00 (0.00–3.25)	0.50 (0.00–2.00)	2.00 (0.00–4.50)	0.191
Female	2.50 (0.00–12.00)	2.00 (0.00–13.00)	3.00 (0.00–12.00)	0.425
*p*-Value	0.007	0.027	0.180	
	**BURNOUT-DP (α = 0.656)**			
	0.00 (0.00–1.00)	0.00 (0.00–1.00)	1.00 (0.00–2.00)	0.024
Sex				
Male	0.00 (0.00–1.00)	0.00 (0.00–0.75)	0.00 (0.00–1.00)	0.498
Female	0.00 (0.00–1.75)	0.00 (0.00–1.00)	1.00 (0.00–2.00)	0.043
*p*-Value	0.091	0.395	0.180	
	**BURNOUT-PA (α = 0.704)**			
	38.00 (35.00–41.00)	37.00 (34.00–40.50)	38.00 (35.50–41.00)	0.245
Sex				
Male	37.50 (34.75–39.25)	36.00 (34.00–38.75)	38.00 (36.75–41.00)	0.023
Female	38.00 (35.00–41.00)	38.00 (34.50–41.50)	38.00 (35.00–41.00)	0.880
*p*-Value	0.391	0.147	0.359	

Values are medians and interquartile ranges (in brackets). R1 = first year of residency, R2 = second year of residency.

Correlation analyses (Table 5) revealed a positive and statistically significant association (*p* < 0.05) between the total burnout score and most of the variables studied. However, notable exceptions were observed in the attitudinal dimensions towards patients with mental illness: specifically, a negative and significant correlation (*p* < 0.05) was identified with the factors of ‘authoritarianism’ and ‘social restrictiveness’. Furthermore, the dimension of ‘putting oneself in the patient’s place’ (empathy) did not show a significant correlation with total burnout (*p* > 0.05). This pattern of results was maintained for the ‘emotional exhaustion’ subscale (a dimension of burnout). The only differences were that, for this particular dimension, neither age nor the dimension of ‘personal fulfilment’ (another burnout subscale) showed a significant correlation (*p* > 0.05) with emotional exhaustion.

In contrast to previous findings, the “depersonalisation” dimension of burnout showed a predominantly positive and significant association (*p* < 0.05) with most of the variables included in the study. However, there were notable exceptions to this pattern, including a single significant negative correlation (*p* < 0.05) with the ‘social restrictiveness’ dimension within attitudes towards patients with mental illness. Furthermore, no significant correlations (*p* > 0.05) were found with several key variables. Specifically, no association was found with participant age, the ‘personal accomplishment’ dimension of burnout, or the empathic subscale of ‘putting oneself in the patient’s place.’ Finally, depersonalisation also did not correlate significantly with the total score for attitudes towards patients or with its subdimensions of “authoritarianism” and ‘benevolence.’

The “personal accomplishment” dimension of burnout showed positive correlations (*p* < 0.05) with empathy and its dimensions, total burnout and the CMH dimension of attitudes towards the mentally ill patients and a negative correlation (*p* < 0.05) with the “social restrictiveness” and “authoritarianism” dimensions of attitudes towards the mentally ill patients (Table 5).

No significant correlation (*p* > 0.05) was observed between “personal accomplishment” and age, Total CAMI, “benevolence”, “emotional exhaustion and depersonalisation”.

### 3.3. Attitudes Towards Mentally Ill Patient

Table 4 presents the results for overall community attitudes toward mental illness and their corresponding dimensions. A statistically significant increase (*p* < 0.05) was observed over the years in the attitudes towards the patients with mental disorders in general, as well as in its dimensions “benevolence” and “community mental health ideology”. On the other hand, a statistically significant decrease (*p* < 0.05) was observed in the dimensions “authoritarianism” and “social restrictiveness”.

When comparing by sex, both “authoritarianism” and “community mental health ideology” dimensions showed significant differences (*p* < 0.05) in general, being higher in the second year in men for “authoritarianism” and in the second year in women in relation to “community mental health ideology”. However, both attitudes towards the mentally ill patient in general and the dimensions “benevolence” and “social restrictiveness” did not differ between the sexes.

Regarding the correlations between attitudes toward patients with mental illness and their corresponding dimensions and the remaining study variables (Table 5), a positive correlation was observed (*p* < 0.05) between attitudes in general with its dimensions “authoritarianism”, “benevolence” and “community mental health ideology”, age, empathy and its dimensions “Standing in the Patient’s Shoes” and “perspective taking”, burnout and its dimension “emotional exhaustion”. Furthermore, the dimension “authoritarianism” showed only positive correlation (*p* < 0.05) with attitudes towards the mentally ill patient in general and its dimension “social restrictiveness”.

**Table 4 nursrep-15-00381-t004:** Association between attitudes towards the patient with mental disorders and its dimensions, sex and years of residence of mental health nurse residents in Spain.

		CAMI		
Variable	Whole Sample	R1	R2	*p*-Value
	125.00 (123.75–128.00)	125.00 (122.00–127.00)	126.00 (124.00–128.00)	0.004
Sex				
Male	125.00 (124.00–127.00)	125.00 (124.00–126.00)	126.00 (125.00–128.00)	0.020
Female	126.00 (123.00–128.00)	125.00 (122.00–128.00)	127.00 (124.00–128.00)	0.028
*p*-Value	0.714	1.000	0.868	
	**CAMI-A** (α = 0.786)			
	24.00 (22.00–26.00)	25.00 (23.00–27.00)	23.00 (21.00–24.50)	0.000
Sex				
Male	25.00 (23.75–26.00)	26.00 (24.25–26.75)	24.00 (22.00–25.00)	0.012
Female	23.00 (22.00–26.00)	25.00 (25.50–27.00)	22.00 (21.00–24.00)	0.000
*p*-Value	0.004	0.312	0.015	
	**CAMI-B** (α = 0.765)			
	43.00 (41.00–45.00)	41.00 (40.00–43.00)	44.00 (43.00–46.00)	0.000
Sex				
Male	42.00 (40.00–44.00)	41.00 (40.00–42.00)	44.00 (43.00–45.00)	0.001
Female	43.00 (41.00–45.00)	42.00 (40.00–43.00)	44.00 (43.00–46.00)	0.000
*p*-Value	0.064	0.355	0.184	
	**CAMI-SR** (α = 0.883)			
	18.00 (16.00–19.00)	19.00 (17.00–20.00)	17.00 (15.00–18.00)	0.000
Sex				
Male	19.00 (17.00–19.00)	19.00 (18.00–20.00)	17.00 (15.75–19.00)	0.024
Female	17.00 (16.00–19.00)	19.00 (17.00–20.50)	16.00 (14.00–18.00)	0.000
*p*-Value	0.058	0.793	0.151	
	**CAMI-CMH** (α = 0.851)			
	41.50 (39.00–44.00)	40.00 (38.00–42.00)	43.00 (41.00–45.50)	0.000
Sex				
Male	40.00 (38.75–42.00)	40.00 (38.00–40.75)	41.00 (39.75–43.25)	0.032
Female	42.00 (40.00–45.00)	40.00 (38.00–43.00)	43.00 (41.00–46.00)	0.000
*p*-Value	0.002	0.266	0.013	

Values are medians and interquartile ranges (in brackets). R1 = first year of residency, R2 = second year of residency.

Significant negative correlations (*p* < 0.05) were found between authoritarianism and all other study variables, except for the depersonalization dimension of burnout, with which no significant association was observed (*p* > 0.05).

In the “benevolence” dimension, a positive correlation (*p* < 0.05) was observed with attitudes towards the mentally ill patient in general and its dimension “community mental health ideology”, age, empathy and all its dimensions, burnout and its dimension “emotional exhaustion”. In addition, negative correlations (*p* < 0.05) were found between the dimension ‘benevolence’ and the dimensions “authoritarianism” and “social restrictiveness”.

Following the same analysis the “social restrictiveness” dimension of attitudes toward patients with mental illness exhibited significant negative correlations (*p* < 0.05) with all other study variables, except for the “authoritarianism” dimension, with which a significant positive correlation was observed. Finally, in the “community mental health ideology” dimension of attitudes towards the patient with mental disorder, a positive correlation (*p* < 0.05) was observed with all the study variables except with the “authoritarianism” and “social restrictiveness” dimensions where the correlation was negative (*p* < 0.05).

**Table 5 nursrep-15-00381-t005:** Spearman’s rank correlation matrix between the study variables in mental health nurse residents in Spain.

VARIABLE	JPSE	JPSE- SPS	JPSE- CC	JPSE- PT	TOTAL MBI	MBI- EE	MBI- DP	MBI- PA	TOTAL CAMI	CAMI-A	CAMI- B	CAMI- SR	CAMI- CMH	AGE
JPSE	1													
JPSE-SPS	0.613 **	1												
JPSE-CC	0.845 **	0.310 **	1											
JPSE-PT	0.920 **	0.575 **	0.629 **	1										
TOTAL MBI	0.496 **	0.070	0.578 **	0.387 **	1									
MBI-EE	0.481 **	0.029	0.570 **	0.388 **	0.793 **	1								
MBI-DP	0.272 **	−0.089	0.360 **	0.201 **	0.623 **	0.633 **	1							
MBI-PA	0.237 **	0.253 **	0.134 *	0.252 **	0.468 **	−0.008	−0.043	1						
TOTAL CAMI	0.226 **	0.252 **	0.126	0.205 **	0.207 **	0.154 *	0.128	0.113	1					
CAMI-A	−0.555 **	−0.411 **	−0.417 **	−0.537 **	−0.288 **	−0.252 **	−0.120	−0.142 *	0.207 **	1				
CAMI-B	0.525 **	0.459 **	0.351 **	0.520 **	0.269 **	0.257 **	0.128	−0.069	0.566 **	−0.599 **	1			
CAMI-SR	−0.644 **	−0.450 **	−0.532 **	−0.585 **	−0.331 **	−0.289 **	−0.203 **	−0.198 **	−0.041	0.635 **	−0.526 **	1		
CAMI-CMH	0.676 **	0.490 **	0.538 **	0.615 **	0.392 **	0.335 **	0.197 **	0.204 **	0.418 **	−0.725 **	0.628 **	−0.780 **	1	
AGE	0.290 **	0.218 **	0.161 *	0.333 **	0.160 *	0.073	0.116	0.131	0.144 *	−0.244 **	0.245 **	−0.198 **	0.275 **	1

* indicates a significant correlation, *p* < 0.05; ** indicates a significant correlation, *p* < 0.01.

## 4. Discussion

This study found moderate to high levels of empathy among Spanish mental health nursing residents, average levels of burnout, and generally positive attitudes towards patients with mental disorders. Empathy tended to increase between the first and second years of residency, while emotional exhaustion was higher among first-year residents. Attitudes also improved slightly with training, showing less authoritarianism and greater benevolence.

These results suggest that the development of empathy during the residency period may contribute to more positive and less stigmatising attitudes towards people with mental disorders.

These findings provide a deeper understanding of how empathy, burnout, and attitudes towards people with mental disorders are interrelated during the specialist mental health training of nurse residents.

The results regarding the levels of empathy observed in our sample are consistent with previous studies suggesting that clinical environments focused on comprehensive care for the individual favour the development of empathetic attitudes [40]. However, the literature sometimes presents inconsistent results regarding the evolution of empathy over time in healthcare professionals, reflecting its dynamic nature influenced by continuous clinical practice and sociocultural expectations linked to gender roles [30].

Contrary to the protective role usually attributed to empathy regarding burnout, our findings reveal a positive association between levels of empathy and dimensions of professional exhaustion. This phenomenon can be understood through the concepts of empathic distress and compassion fatigue, where a high degree of empathic involvement without adequate mechanisms of emotional regulation or coping increases vulnerability to burnout [18]. In this regard, resilience mechanisms such as mindfulness practice, coping strategies, and organisational support are key elements in mitigating emotional exhaustion and promoting professional sustainability [41,42]. Our study highlights the need to incorporate these resilience-building components into specialised mental health nursing training to optimise professional well-being and quality of care.

Furthermore, an integrative interpretation of the three main constructs—empathy, burnout, and attitudes—was conducted to highlight their potential interrelationship. Empathy may function as a mediating or moderating factor between burnout and attitudes towards patients, while positive attitudes appear to buffer emotional exhaustion and foster professional resilience.

The most relevant findings of the study are discussed below.

### 4.1. Empathy

In line with other research, our results showed that empathy of mental health nursing residents, measured globally, was relatively high [43] probably due to the specificity of a people-oriented work environment [44].

Despite this, Villalba-Arias et al. [30] suggest that more studies are needed to be able to draw more general trend results. On the other hand, the existing literature shows mixed results on whether empathy increases or decreases over time in the health professions trajectory. Ferri et al. found that empathy decreases across studies. However, during mental health residency, empathy showed an increase over time in agreement with the findings of other authors such as Meza et al.; Ghaedi et al.; Bogiatzaki et al. and Yuguero et al. [16,20,44,45,46].

Specifically, our findings show an increase in scores on all dimensions of empathy between the first and second year of residency in the mental health speciality. This result could indicate, as has already been pointed out by other authors, a progressive development of empathic competencies throughout the training process [35]. The increase in empathy observed during residency could be related to greater curricular exposure to mental healthcare, opportunities for reflective practice, and the personal maturation of residents as they consolidate their professional identity.

In relation to gender differences, the existing literature consistently indicates higher scores in women for global empathy [22,23,24], a trend that is also reflected in our data, especially in the last year of residency. However, when analysing the specific dimension of ‘putting oneself in the patient’s place’, no significant differences were observed between men and women, which suggests that this skill is developed in a common and equal manner in both sexes [25].

On the other hand, in the perspective-taking dimension, our results show higher scores for women during part of the training period, although this difference disappears in the second year of residency, where a balance is reached between both sexes. We believe that this finding could be more related to the characteristics of the training process—based —on continuous clinical practice, direct exposure to patients with mental disorders, theoretical teaching and analysis of clinical cases—than to the sex of the residents [47].

In this sense, and in line with other research, the residency could be configured as the ideal setting that challenges residents to practice and hone their empathic skills, improving their ability to understand diverse perspectives [48,49].

Finally, the way of experiencing patients’ emotions, included in the dimension of ‘compassionate care’, is again of particular relevance in the female sex, differentiating it from the male sex, as its score increases throughout the entire professional trajectory during the residency. Some authors suggest that the differences observed in both general empathy and its dimensions may be largely due to cultural expectations associated with gender roles [25], where women are attributed a role as primary caregivers with a predominant affective component, which would explain their high scores in the compassionate care dimension.

Furthermore, empathy is directly related to positive attitudes towards people with mental disorders, which contributes to reducing authoritarianism and social restrictiveness towards this type of users. These findings are in line with what has been reported in the international literature [21,22,50].

Although empathy appears to play a significant role in the relationship between burnout and attitudes, no formal mediation analysis was performed; therefore, the results should be interpreted with caution, considering that empathy could function as a potential mediator rather than assuming a direct mediating effect.

### 4.2. Burnout

Although the scientific literature has consistently identified empathy as a protective factor against burnout syndrome [18], our study revealed a contrasting pattern. We observed a significant direct relationship: burnout levels increased progressively with overall empathy scores and its various dimensions. This trend may suggest that, in certain emotionally demanding contexts, higher levels of empathy could paradoxically increase vulnerability to burnout. Furthermore, a significant increase in burnout was observed over the years of residency among male participants, which may indicate that the defence mechanisms employed by males are less effective in coping with prolonged work-related stress [29].

In relation to the differences that burnout may present according to sex, in our study statistically significant differences were only observed in first-year residents, with women presenting higher levels of professional burnout. The dimension of emotional exhaustion, recognised as the first to manifest itself within the burnout syndrome, showed significantly lower scores in men during the first year of residency. However, this trend is reversed in later stages, with specialists having the highest scores on this dimension. These results are consistent with the findings of the systematic review by López-López et al. [51] who conclude that men have fewer resources to cope with work-related stress, which could lead to an earlier onset of burnout.

On the other hand, Ferri et al. [16] argue that the depersonalisation dimension is associated with the affective distancing of the professional from the patient, and may represent a defensive mechanism of emotional protection. Although several studies, such as that of Ramírez et al. [52], indicate that the male sex is associated with a greater predisposition to depersonalisation, our results show the opposite trend: women showed a statistically significant increase in this dimension after their residency. However, this increase did not reach levels of statistical significance. Some authors have proposed that this difference could be related to the high proportion of women in mental health nursing teaching units [31].

In addition, the humanistic approach present in training programmes in health sciences—and particularly in nursing studies—which promotes a view of the patient as a human being endowed with emotions and capable of suffering, could be influencing the low levels of depersonalisation detected during residency. This phenomenon may be explained by the persistence of a pedagogical model based on direct teaching and professional accompaniment through clinical and teaching tutors [32].

Collectively, these findings underscore the need for future research aimed at a more in-depth analysis of the observed differences across the dimensions of burnout, with particular attention to sex as a potential moderating variable.

The scientific literature indicates that nurses maintain stability in their personal accomplishment while they are active in the care work [28]. However, during the research, a significant increase was observed in the samples of men who completed the mental health residency. However, further studies should be conducted with male mental health nursing residents in order to show conclusions to these findings.

Furthermore, burnout and its dimensions of emotional exhaustion and personal accomplishment are related to a decrease in authoritarianism and social restrictiveness in the attitudes of the residents towards this type of patients. This finding is also reflected in the literature, as nurses in residency training who report a sense of accomplishment and increased knowledge in managing patients with mental disorders tend to refrain from authoritarian behaviours and social isolation [21,22,50].

From the specialised health literature, we observe that, despite the fact that the training programme for the specialty of mental health nursing published in the BOE number 123 of 24 May 2001, which approves this programme, indicates that “the resident will show an open and non-discriminatory attitude, as well as fight against the stigma suffered by patients with mental disorders”, there are no studies that evaluate these attitudes in Spanish residents.

In our study, in the dimension “authoritarianism”, higher scores were observed in male students. It is possible that in them there are still paternalistic behaviours typical of the stigmatisation towards mental health patients [53] being probable that they have acquired them from the professionals of the units [54]. These paternalistic attitudes seek to decide for the patient and direct them in the best possible way, taking little into account their opinion and seeking to protect them. With the passage of time these behaviours will decrease and it will be in the female sex in which these dimensions predominate [29]. In spite of this and in general, in mental health residents “authoritarianism” and “social restrictiveness” decrease throughout the two-year stay in both sexes, a fact that seems to indicate an assimilation of the patient with mental disorder as an adult person, self-sufficient and without danger to others.

In the dimension “benevolence” and “community mental health ideology” it can be observed that, in these two dimensions, the female sex starts from a higher score, and there are significant changes throughout the residency in all participants who improve their scores [55].

It is observed that a higher empathy score significantly improves general attitudes towards the patient with mental disorder, increasing the latter as in the study by Jung et al. [33].

Finally, the reliability of the scales was good, with the exception of the total score of the JSP-SPS and MBI-DP, as also shown by the analyses performed by Díaz-Narváez et al. [37] who, despite the low figures, consider them to be good scores. Current studies recommend that, in the Jefferson scale, only the reliability of the total score should be calculated, without dividing it into dimensions [56]. Additionally, the low score on “depersonalization” can also be observed in the study of Guido et al. and they report that low scores are acceptable. Scores similar to the reliability scores of our studies are also observed in other studies, especially in non-English speaking populations [39,57,58].

The conclusions of this study transcend the educational sphere and offer valuable insights for nursing staff management and institutional policy-making. The results highlight the need for healthcare organisations to implement strategies aimed at ensuring sustainable working environments, where an appropriate balance is maintained in care workloads and emotional support programmes are promoted to prevent burnout syndrome. It is also proposed to strengthen continuing education and mentoring, promoting the development of professional empathy without affecting the psychological health of teams. In the case of Spain, it is advisable to incorporate the cultural dimension into the design of mental health and resilience initiatives, adjusting them to the social and organisational specificities of the healthcare system. Finally, these conclusions open up avenues for future research focused on analysing the factors involved in empathy, emotional fatigue and stigma, and assessing the impact of strategies aimed at consolidating institutional resilience and work commitment among healthcare professionals.

The present study has several limitations that should be considered when interpreting the results. Firstly, the cross-sectional design does not allow causal relationships to be established between the variables analysed, limiting the conclusions to associations observed at a single point in time. Secondly, the non-probabilistic sampling method employed does not guarantee the representativeness of the general population of mental health nursing residents, which may affect the generalisability of the findings. Furthermore, although participants were recruited from different teaching units of the Spanish National Health System, potential heterogeneity among residency programmes—such as differences in training structure, workload, and institutional support—could have influenced empathy, burnout, and attitudes. Finally, levels of empathy, burnout, and attitudes towards people with mental disorders were assessed using self-administered questionnaires, which implies a possible social desirability bias, as participants may have responded according to what they considered socially acceptable or expected, rather than reflecting their true perceptions or experiences. In addition, the use of self-report measures for all variables at a single time point introduces the risk of common method bias. Finally, future studies should explore the possibility of bidirectional relationships: while empathy may promote more positive attitudes and reduce burnout, maintaining benevolent attitudes toward patients could, in turn, reinforce empathic engagement and emotional well-being. Longitudinal and mixed-method designs would help clarify these dynamic interactions.

## 5. Conclusions

Mental health nurse residents in Spain displayed moderately high empathy, medium levels of burnout, and overall positive attitudes toward patients with mental disorders. No significant correlation was found between empathy and burnout, although empathy was positively associated with supportive attitudes toward these patients. These results partially support the study hypothesis, suggesting that while empathy and burnout may not be directly linked, greater empathy contributes to more positive and less stigmatising attitudes.

The findings highlight the importance of incorporating emotional regulation, reflective practice, and resilience-building strategies within mental health nursing training programmes. Strengthening these competencies could help sustain empathy and prevent emotional exhaustion, ultimately improving the quality of therapeutic relationships and patient care.

This research contributes to the knowledge about the psychosocial processes that take place during specialised healthcare training, providing relevant evidence for the design of interventions that promote emotional well-being in highly demanding clinical contexts. The findings allow us to advance the understanding of the role played by attitudinal and emotional variables—such as empathy and burnout—in the construction of a more humanised professional practice.

This study can serve as a basis for the development of training programmes aimed at strengthening emotional competencies from a holistic health perspective, sensitive to gender differences and the impact of organisational contexts. It also offers implications for the design of institutional policies that integrate professional care as a key element in the quality of mental healthcare.

On a broader level, it is hoped that these results will contribute to the promotion of interdisciplinary strategies between psychology, nursing, and education to consolidate more sustainable, safe, and mental health-oriented clinical and training environments for both patients and professionals in training.

## Data Availability

The data presented in this study are available on request from the corresponding author. The data are not publicly available due to privacy.

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
