# Peer review of "Empathy, Burnout, and Attitudes Toward Patients with Mental Disorders Among Mental Health Nurse Residents in Spain: A Cross-Sectional Study"

_nursrep, 2025, doi:10.3390/nursrep15110381_

Round 1

Reviewer 1 Report

Comments and Suggestions for Authors

It is my pleasure to review the manuscript entitled “Empathy, Burnout, and Attitudes toward Patients with Mental Disorders among Mental Health Nurse Residents” by Daniel Román-Sanchez et al. (2025). This study explores the interrelationship between empathy, burnout, and attitudes toward patients with mental disorders among nurse residents in Spain. The topic is highly relevant and timely, given the growing concern about the psychological well-being of mental health professionals and its impact on patient care. Below is a comprehensive and deeply analytical critique of the article, structured according to the requested criteria. Below is a detailed critique organized by key manuscript sections.

My comments are as follows:

  • Title: The preposition "on" in "Practice on Research and Evidence-Informed Practice” is grammatically awkward. A more conventional and accurate phrasing would be: “Knowledge, Attitude, and Practice Regarding Research and Evidence-Informed Practice…” or “…Toward Research and Evidence-Informed Practice…” the title could benefit from a slight hint at the methodological approach (e.g., “A Cross-Sectional Survey” or “A Descriptive Study”), which would enhance scientific transparency and appeal to systematic reviewers. “Empathy, Burnout, and Attitudes toward Patients with Mental Disorders among Mental Health Nurse Residents in Spain: A Cross-Sectional Study.”
  • Abstract: The abstract provides a general overview, including the research problem, sample size, methodology, findings, and recommendations. However, Need to add age and gender in results. The abstract omits essential quantitative details such as correlation coefficients, p-values, or reliability indices. Adding a geographic keyword such as “Spain”
  • Introduction: Although the relevance of empathy and burnout is well presented, the research problem is implied rather than explicitly stated. Certain portions include overly broad or repetitive statements about empathy’s importance or burnout’s prevalence, which could be condensed. A sharper focus on the interaction among the three variables would maintain reader engagement and reduce redundancy. The introduction lacks discussion on Spanish cultural and educational factors that might influence empathy and attitudes (e.g., national residency structure, mental health stigma in Spain). The introduction could benefit from a brief mention of how this study aligns with global nursing initiatives promoting compassion, such as the WHO’s mental health action framework or the ICN’s nursing education standards. Many cited studies are summarized rather than critically evaluated to identify contradictions, methodological gaps, or limitations in prior research.
  • Methods

-Study design and participants: The authors did not provide a power analysis or sample size calculation to determine the adequacy of the 214 participants for detecting significant correlations. A priori power analysis (e.g., using G*Power) would have demonstrated methodological rigor and statistical validity. The authors do not specify how many participants came from each area or whether institutional characteristics (e.g., training curriculum, clinical load) varied by site. No mention is made of stratification by year of residency, although results later compare first- and second-year residents. While the authors mention that participants were current residents in the mental health specialization, they did not specify exclusion criteria (e.g., residents on leave, with prior psychiatric specialization, or incomplete participation). The article refers to the study being conducted across "teaching units of the Spanish National Health System," yet it does not describe the institutional environments (hospital type, region, or training structure differences). Variations between training sites could influence empathy and burnout levels, and this heterogeneity remains unaddressed.

-Instruments and Variables: Empathy: need to report number of items for each dimension. Burnout: need to report number of items for each dimension. Attitudes towards mentally ill patients: It is not fully explained whether the authors employed the full 40-item CAMI or a shorter version. Need citation of interpretation and internal consistency.

- Statistical Analysis: Although the authors mention non-normality, they do not report tests for homogeneity of variance (Levene’s test) or independence of observations. The manuscript does not describe how missing or incomplete responses were treated—whether through deletion, imputation, or pairwise exclusion.

- Ethical Considerations: The manuscript contains contradictory statements about ethical review. In one section, the authors claim approval was obtained from the Research Ethics Committee of Cádiz, while elsewhere they state that ethical approval was not necessary due to the study’s minimal-risk nature. Although the authors mention that consent was obtained electronically, they do not specify the mechanism (e.g., digital checkbox, information sheet preceding the survey). Best ethical practice requires detailing how consent was documented, what information participants received about data use, withdrawal rights, or data retention duration. There is no description of how long data will be stored, who has access, or whether it will be destroyed after a set period. It is not stated whether the study received institutional permission from participating mental health residency programs or hospital administrators. Were files password-protected? Who had access? And need permission to use tool.

Data collection: not mention. the process of data collection is not described. need the Who conducted the data collection?

  • Results: While tables are clear and well-labeled, the text often repeats rather than interprets tabulated data. The Results section should balance statistical reporting with brief interpretive summaries. Although the median and IQR values are mentioned, the authors do not provide exact numeric data for each empathy subdimension (e.g., JSE total = 112, IQR = 105–119).
  • Discussion: The discussion briefly acknowledges discrepancies with prior studies but does not critically analyze why differences occurred (e.g., cultural, contextual, or methodological variations). The authors do not adequately situate the results within Spanish sociocultural and healthcare contexts. The discussion touches on burnout but does not explore the psychological resilience mechanisms that might protect nurse residents, such as coping skills, mindfulness, or organizational support. The three major constructs—empathy, burnout, and attitudes—are discussed separately rather than interdependently. The authors miss the opportunity to explain how empathy may mediate or moderate the relationship between burnout and attitudes, or how positive attitudes could buffer emotional exhaustion. This gap limits the conceptual richness of the discussion. The absence of correlation between empathy and burnout is briefly mentioned but not deeply analyzed. The authors could have explored this paradox in light of empathic distress, emotional regulation, or compassion fatigue theories, which propose that empathy, without adequate coping mechanisms. Some parts of the discussion reiterate statistical results (e.g., median values and significance levels) rather than focusing on interpretation. While limitations are listed, the section lacks depth of discussion. The sampling process is not critically evaluated. All variables were measured using self-report instruments at a single point in time, which raises the possibility of common method bias.
  • The implication focuses solely on education and omits implications for nursing management and policy—for instance, how hospital administrators could develop supportive environments, improve staffing ratios, or integrate emotional well-being programs to prevent burnout. The authors do not address how findings might inform culturally sensitive mental health training in Spain or similar contexts. Although burnout was a central variable, the section fails to propose implications for organizational resilience-building, stress management training, or mentorship programs that can sustain empathy while reducing emotional exhaustion. The authors miss the opportunity to frame implications for future studies—for example, exploring mediating or moderating effects among variables, or evaluating intervention programs designed to enhance empathy and reduce stigma.

Author Response

Dear Reviewer,

We would like to sincerely thank you for the time and dedication you devoted to reviewing our manuscript. Your thoughtful and constructive feedback has undoubtedly contributed to improving the quality and clarity of our work.

Please find attached our detailed point-by-point responses to all your comments.

With our best regards,

Reviewer 2 Report

Comments and Suggestions for Authors

Dear authors,

This article is particularly interesting because it addresses a crucial topic in the training and well-being of mental health professionals: the relationship between empathy, burnout, and attitudes toward patients with mental disorders.

My suggestions and comments on the article are as follows:

The title clearly indicates the topic of the article, making it easy to search the database.

The abstract is well-structured and provides a clear, comprehensive overview of the study.

The introduction broadly addresses current issues regarding the research topic: the importance of empathy for the work of nursing professionals, and burnout as a major stress factor in the care of people with mental disorders.

The hypothesis and objectives of the study are clearly defined.

The methodology is appropriate for addressing the study's objectives and is described in detail to allow for replication of the study. However, I believe the dimensions analyzed by the Jefferson Empathy Questionnaire should be indicated or that it is a one-dimensional questionnaire, as this is not clear.

The results are presented in detail with tables that facilitate their understanding, although:

- I believe that the McDonald's omega test would have been more appropriate to analyze the internal consistency of the CAMI questionnaire, since Cronbach's alpha underestimates the results on Likert-type scales with five response options.

- The statistical measures used for the detailed analysis of the items should be indicated.

- A table should be compiled with the psychometric data from the analyzed questionnaires to facilitate the reader's understanding.

The discussion provides a detailed and accurate analysis of the results obtained and establishes relationships with previous studies that address this research topic. A detailed analysis of the study's limitations is also provided, which may help readers draw their own conclusions about the results obtained.

I believe that the conclusions indicate the importance and implications of the study, but do not clearly address the hypothesis and objectives proposed with the results obtained. What levels of empathy, burnout, and attitudes toward patients with mental health problems do resident mental health nurses display? Do residents with greater empathy have lower levels of burnout?

The references are numerous and relevant to the topic under study.

Kind regards.

Author Response

(The authors gave the same response as above.)

Round 2

Reviewer 1 Report

Comments and Suggestions for Authors

It is my pleasure to review the manuscript entitled “Empathy, Burnout, and Attitudes toward Patients with Mental Disorders among Mental Health Nurse Residents in Spain: A Cross-Sectional Study.” In this article, the authors investigated the interrelationship among empathy, burnout, and attitudes toward patients with mental disorders in a sample of 214 Spanish mental health nurse residents. The study addresses a relevant issue in nursing education and practice—how emotional competencies such as empathy interact with occupational well-being and stigma. By exploring these relationships within a culturally specific residency context, the paper contributes to understanding the emotional landscape of future mental health professionals. However, while the study is timely and methodologically sound, several aspects require critical reflection.

My comments are as follows:

  • Introduction: while the introduction flows logically, the specific research problem—that is, the lack of studies exploring empathy, burnout, and stigma simultaneously among Spanish mental health nursing residents—is not articulated in a single, explicit statement. while the literature review is extensive, it could be better synthesized to identify the knowledge gap. Currently, the text summarizes findings from multiple studies but does not clearly indicate what is missing in the current body of evidence.
  • Methods

-Instruments and Variables: Attitudes towards mentally ill patients: Need citation of interpretation and internal consistency.

- Ethical Considerations: The manuscript contains contradictory statements about ethical review. In one section, the authors claim approval was obtained from the Research Ethics Committee of Cádiz, while elsewhere they state that ethical approval was not necessary due to the study’s minimal-risk nature. Although the authors mention that consent was obtained electronically, they do not specify the mechanism (e.g., digital checkbox, information sheet preceding the survey). Best ethical practice requires detailing how consent was documented, what information participants received about data use, withdrawal rights, or data retention duration. There is no description of how long data will be stored, who has access, or whether it will be destroyed after a set period. It is not stated whether the study received institutional permission from participating mental health residency programs or hospital administrators. Were files password-protected? Who had access? And need permission to use tool.

Data collection: not mentioned subheading. the process of data collection is not described. need the Who conducted the data collection?

  • Discussion: the discussion sometimes merges results interpretation with literature review, which dilutes the analytical strength of the argument. A clearer division between “what was found” and “what it means” would enhance readability and conceptual precision. some comparisons remain descriptive rather than analytical. For instance, when citing studies showing empathy increases during training, the authors could have explored why this occurs — possibly due to curriculum exposure, reflective practice, or maturation. Similarly, they mention that attitudes improve with empathy but could have connected this to attitudinal change theories (e.g., contact hypothesis, cognitive–affective models). because no mediation analysis was actually conducted, the authors’ claims should have been phrased more cautiously (e.g., “Empathy may function as a potential mediator” rather than “Empathy mediates”). Additionally, discussing possible bidirectional effects.

Author Response

We sincerely thank the reviewer for their time, careful reading, and constructive feedback. All comments and suggestions have been carefully considered and addressed in the revised version of the manuscript, as detailed in the attached point-by-point response document. We believe that these revisions have significantly improved the clarity, methodological rigor, and overall quality of the paper.

Reviewer 2 Report

Comments and Suggestions for Authors

Dear authors,
I have no further comments or suggestions to make.
Kind regards.

Author Response

We sincerely thank the reviewer for their time, careful evaluation, and positive feedback. We truly appreciate their consideration and are pleased that the revised version of the manuscript meets their expectations.